# Bispecific Antibodies: A Review of Development, Clinical Efficacy and Toxicity in B-Cell Lymphomas

**DOI:** 10.3390/jpm11050355

**Published:** 2021-04-29

**Authors:** Ross Salvaris, Jeremy Ong, Gareth P. Gregory

**Affiliations:** 1Monash Haematology, Monash Health, Clayton, VIC 3168, Australia; jeremy.ong@monashhealth.org (J.O.); gareth.gregory@monashhealth.org (G.P.G.); 2School of Clinical Sciences at Monash Health, Monash University, Clayton, VIC 3168, Australia

**Keywords:** bispecific antibodies, immune cell therapy, B-cell lymphoma, diffuse large B-cell lymphoma (DLBCL), cytokine release syndrome, neurotoxicity, tumour flare

## Abstract

The treatment landscape of B-cell lymphomas is evolving with the advent of novel agents including immune and cellular therapies. Bispecific antibodies (bsAbs) are molecules that recognise two different antigens and are used to engage effector cells, such as T-cells, to kill malignant B-cells. Several bispecific antibodies have entered early phase clinical development since the approval of the CD19/CD3 bispecific antibody, blinatumomab, for relapsed/refractory acute lymphoblastic leukaemia. Novel bsAbs include CD20/CD3 antibodies that are being investigated in both aggressive and indolent non-Hodgkin lymphoma with encouraging overall response rates including complete remissions. These results are seen even in heavily pre-treated patient populations such as those who have relapsed after chimeric antigen receptor T-cell therapy. Potential toxicities include cytokine release syndrome, neurotoxicity and tumour flare, with a number of strategies existing to mitigate these risks. Here, we review the development of bsAbs, their mechanism of action and the different types of bsAbs and how they differ in structure. We will present the currently available data from clinical trials regarding response rates, progression free survival and outcomes across a range of non-Hodgkin lymphoma subtypes. Finally, we will discuss the key toxicities of bsAbs, their rates and management of these adverse events.

## 1. Introduction

In the last decade, novel therapies have emerged that can achieve complete remissions with durable response in patients with heavily pre-treated, relapsed/refractory (R/R) B-cell lymphomas, such as diffuse large B-cell lymphoma (DLBCL) and follicular lymphoma (FL). Immune cellular therapies such as chimeric antigen receptor (CAR) T-cell therapy and bispecific antibodies (bsAbs) have provided specificity to immuno-oncology with different methods to engage the patient’s own T-cells to kill malignant B-cells. 

Ongoing bsAb trials have expanded beyond the relapsed/refractory setting. In the upfront DLBCL setting, bsAbs are being combined with R-CHOP chemotherapy (rituximab, cyclophosphamide, doxorubicin, vincristine, prednisolone) as well as with agents such as polatuzumab vedotin, lenalidomide and atezolizumab. These strategies may lead to combination therapies that obviate the need for initial intensive chemotherapy in some non-Hodgkin lymphoma (NHL) subtypes. In an elderly population with co-morbidities, this may become an attractive alternative to the current standard of care of chemotherapy. However, bsAbs may present additional overlapping toxicities in addition to unique challenges, such as cytokine release syndrome (CRS), which must be mitigated to maximise their clinical efficacy.

This review aims to highlight the different bsAbs currently in advanced stage clinical development for B-cell lymphomas. We will present their biology and mechanism of action, clinical efficacy data available to date and outline common toxicities and their management. Familiarity with these agents is important as they potentially represent a new drug class with a range of indications in the treatment of B-cell lymphomas. However, their treatment protocols can be complex and we aim to illustrate strategies to mitigate and manage key toxicities such as CRS.

## 2. Development and Mechanism of Activity

### 2.1. Historical Development

Bispecific antibodies have only recently become approved for use in haematological conditions with the clinical translation and efficacy of blinatumomab for R/R acute lymphoblastic leukaemia (ALL) and emicizumab for haemophilia A. However, the origin of the technology to create bsAbs can be traced back 50 years earlier when bispecific antigen binding fragments were successfully coupled and demonstrated agglutination of two different cell lines in vitro [1]. The first bsAb approved for therapeutic use was catumaxomab (CD3/epithelial cell adhesion molecule) in 2009 for intraperitoneal treatment of malignant ascites. However, fatal hepatoxicity with intravenous administration limited the therapeutic scope of this agent and it was withdrawn a decade later [2]. The marked clinical activity of blinatumomab (CD3/CD19) for the treatment of R/R ALL and its subsequent approval has led to a surge in the development of bsAbs and their clinical translation in the treatment of lymphomas [3].

### 2.2. Mechanism and Structure

While there are in excess of 100 academic and industry-derived molecules in clinical development, many with similar antigen/receptor targets, there are vast differences between the molecules in their manufacturing and structure. There are mechanistic differences in the activity of bsAbs, with the majority in advanced clinical development being designed to facilitate T-cell engagement and activation of antibody-dependent cellular cytotoxicity (ADCC) against the tumour cell ultimately via perforin/granzyme-mediated cell death. This is achieved through direct binding of CD3ε of the T-cell receptor complex (bypassing T-cell major histocompatibility complex specificity) to a cell surface tumour antigen on the target cell [4]. Other mechanisms of action of bsAbs include cell surface receptor activation or inhibition (in cis on the surface of a single target cell), facilitating co-factor positioning to enhance activation/function (as is the mechanism of emicizumab via factor IXa/factor X for haemophilia A) and chaperoning as a carrier protein for transmembrane transporting to facilitate activity of its cargo. As these other mechanisms are not currently advanced in clinical development for lymphoma, their mechanism will not be discussed further in this review [4,5].

### 2.3. Bridging Effector Cells to Target Cells

#### 2.3.1. Fragment Based bsAb

Similar to naturally occurring antibodies, bsAbs involve permutations of heavy and light chain variable domains, with or without retention of the fragment crystallisable region (Fc) domain. The Fc domain is the tail region of the antibody which interacts with cell surface receptors, Fc receptors. The antigen-binding fragment (Fab) is the part of the antibody that binds to the antigen. As a naturally occurring antibody contains two heavy and two light chain variable regions, there can be 16 potential recombinations resulting in significant antibody diversity during manufacturing. One method to overcome this is to generate individual antigen-binding fragments (Fabs) and fuse them via chemical coupling such as oxidation or physical linkage through addition of linker proteins. Blinatumomab is an example of a fragment based bsAb, wherein antibody fragments are combined without the presence of the Fc domain. While this structure is easier and cheaper to manufacture, the absence of the Fc region is the reason that the molecule undergoes rapid degradation. These unfavourable pharmacokinetics explain its requirement for constant intravenous infusion and also its limited clinical activity in the treatment of NHL as compared with ALL.

#### 2.3.2. Fc Containing bsAb

The majority of bsAbs in advanced stage clinical development for lymphoma are those containing the Fc region, as these most closely resemble naturally occurring IgG antibodies and hence maintain favourable pharmacokinetic properties. This also renders these bsAbs to be least immunogenic, enhancing tolerability and efficacy upon repeat administration. The Fc domain of bsAbs is commonly re-engineered to prevent FcyRs and C1q binding to prevent excessive cytokine release from recruited T cells and to minimise off-target toxicity such as T-cell mediated hepatotoxicity. Preserving FcRn binding prolongs the half-life of the agent in vivo. The knob and hole manufacturing approach involves separately derived half antibodies with “knob” and “hole” mutations in the CH3 domains that facilitate homodimerization [6]. Reduction and oxidization of disulfides within the hinge region allows for disulfide bridge formation to complete the binding of the half antibodies into the bsAb.

Another key difference between certain bsAbs is the number of Fab arms within each antibody. Mosunetuzumab (CD3/CD20; Genentech) has a 1:1 CD3:CD20 ratio of Fab arms, whereas glofitamab (CD3/CD20; Genentech) has a 1:2 CD3:CD20 ratio, facilitating higher avidity of the binding of the T-cell via two CD20 antigens on the target B-cell simultaneously. Notably, the CD20 Fab of glofitamab is derived from the 2nd generation CD20 monoclonal antibody, obinutuzumab, which itself was glycoengineered to enhance binding and increase ADCC compared to first generation monoclonal CD20 Abs such as rituximab. It is hypothesised that the differential avidity of the antibodies due to the differing CD3:CD20 ratios is the reason for possible differential activity of these molecules against aggressive and indolent B-cell histologies, although this observation is subject to further preclinical and clinical investigation.

In contrast to other bsAbs based around an IgG backbone, IGM-2323 (CD3/CD20; IGM Biosciences) is a bsAb based on an IgM backbone with grafting of IgG antibodies onto the multimeric IgM frame and addition of a CD3-binding Fv domain [7]. This creates an antibody with 10 binding sites and provides 100× avidity compared to the corresponding IgG antibody, with greater ADCC observed in vitro. Early clinical data also suggests this structure associates with less significant CRS and different response-kinetics.

### 2.4. Non-CD3 Engaging bsAbs

While the majority of molecules in advanced clinical development are CD3/CD20 bsAbs, early phase clinical translation includes a myriad of other effector targets and mechanisms. RG6076 (CD19/4-1BBL; Genentech) is an example of alternative B- and T-cell antigenic targets in order to engage T-cell ADCC. AFM13 (CD16A/CD30) engages NK-cells and macrophages via CD16A to CD30+ malignancies such as CD30+ T-cell lymphomas and Hodgkin lymphoma. Indeed, as the efficiency of design and production has developed, bispecific antibodies can be created with virtually any desired antigen specificities for malignant and non-malignant indications. 

## 3. Clinical Efficacy

### 3.1. Blinatumomab

Of B-cell malignancies, bsAbs were first used to treat R/R B-ALL. Blinatumomab, an anti-CD19/CD3 bispecific antibody, showed efficacy in B-ALL and was approved by the Food and Drug Administration (FDA) for this indication in 2014 [8,9]. On the back of this data, blinatumomab was trialled in patients with NHL.

Blinatumomab has been assessed in R/R DLBCL in three clinical studies [10,11,12]. A phase I study by Goebeler et al. administered blinatumomab as a continuous infusion for four to eight weeks followed by an additional four weeks of therapy if clinical benefit was achieved [10]. Doses below 60 µg/m^2^/day achieved poor response rates and a dose of 90 µg/m^2^/day was limited by neurotoxicity. In the extension phase, 11 patients with R/R DLBCL received the target dose with an overall response rate (ORR) of 55% and a complete remission (CR)/unconfirmed complete remission rate of 36% [10]. 

A phase II study by Viardot et al. assessed blinatumomab in R/R DLBCL [11]. Out of 21 evaluable patients, the ORR was 42% with a CR rate of 19% [11]. Of the five (22%) patients who discontinued stepwise dosing due to adverse events, four did so due to neurological events [11]. A phase II study by Coyle et al. evaluated blinatumomab as a second salvage in R/R DLBCL [12]. In 41 patients, after 12 weeks of therapy, the ORR was 37% with a CR rate of 22% [12]. However, due to the high rate of treatment discontinuation resulting from disease progression, only 59% of patients received more than 80% of their intended dose [12]. Ten patients (24%) experienced a grade 3 or higher adverse neurological event [12]. 

A number of bispecific CD20/CD3 antibodies are now in development for the treatment of both indolent and aggressive NHLs. Early efficacy and safety data has been presented for a number of these agents. These clinical trials are still ongoing however and further follow-up is required.

### 3.2. Glofitamab

In the setting of R/R B-cell NHL, NP30179 (NCT03075696) is an ongoing phase I/Ib trial with glofitamab, a CD20/CD3 antibody comprising two Fab regions for CD20 and one Fab region for CD3. Obinutuzumab is administered before initiating glofitamab in order to occupy CD20 on the surface of lymphoma cells as well as depleting peripheral B-cells to reduce the risk of CRS. 

Data has been presented regarding the phase I dose escalation cohort where glofitamab was administered at a clinically relevant dose of 0.6 to 25 mg. Obinutuzumab pre-treatment (Gpt) was given before intravenous (IV) glofitamab was administered either two or three weekly for up to 12 cycles. There were 118 patients, 102 with aggressive NHL (aNHL) and all indolent NHL (iNHL) patients had FL. In the patients with aNHL, the ORR was 47% and the CR rate was 34% [13]. In the group of patients who received 10 to 25 mg, the ORR was 54% and the CR rate was 42% suggesting response was related to the dose given [13]. For patients in CR, after a median follow-up of 7.1 months, the median duration of CR was not reached [13]. In patients with FL, the ORR was 77% and in patients who received 10 to 16 mg the ORR was 88% [13]. 

Glofitamab was then administered IV in a weekly step-up dosing regimen with a schedule of either 2.5/10/16 mg or 2.5/10/30 mg. Glofitamab was given every three weeks for up to 12 cycles. Interim data for NP30179 presented the results of 38 patients with R/R NHL who received glofitamab step-up dosing with Gpt [14]. There were 17 patients (45%) who received 2.5/10/16 mg and 21 patients (55%) who received 2.5/10/30 mg. Patients with aNHL (73.7%) and iNHL (26.3%) were included. In all efficacy-evaluable patients (*n* = 32), the ORR was 62.5% with a CR rate of 40.6% [14]. For patients with aNHL (DLBCL, transformed follicular lymphoma (tfFL), mantle cell lymphoma (MCL), Richter’s transformation, grade 3B FL), the ORR was 50% with a CR rate of 29.2% [14]. For the eight patients with grade 1-3A FL, the ORR was 100% with a CR rate of 75% [14].

In the R/R NHL setting, glofitamab is also being assessed in a number of ongoing trials. In the phase Ib trial NP39488 (NCT03533283), glofitamab is being combined with either polatuzumab vedotin, a humanised anti-CD79b monoclonal antibody conjugated to monomethyl auristatin E (MMAE), or atezolizumab, an anti-PD-L1 monoclonal antibody. In the phase Ib trial G041943 (NCT04313608), glofitamab or mosunetuzumab is combined with gemcitabine plus oxaliplatin (GemOx) in R/R DLBCL or high-grade B-cell lymphoma. G041944 (NCT04408638) is a phase III trial comparing glofitamab with GemOx versus rituximab with GemOx in patients with R/R DLBCL in the third line or later for transplant-eligible subjects and second line or later in transplant-ineligible subjects.

In the front-line setting, glofitamab is also being studied. A phase Ib trial NP40126 (NCT03467373) is evaluating glofitamab in patients with untreated DLBCL more than 60 years of age with an age-adjusted International Prognostic Index (IPI) of 2–3. Glofitamab will either be added to CHOP in addition to obinutuzumab or rituximab, or it will replace the anti-CD20 antibody in a glofitamab plus CHOP arm.

### 3.3. Mosunetuzumab

Mosunetuzumab (RG7828, RO7030816), a full-length bispecific CD20/CD3 antibody, is being trialled in both the R/R NHL setting as well as first line, either as a single agent or in combination with CHOP-like regimens and polatuzumab vedotin.

In patients with R/R NHL, the phase I/Ib trial GO29781 (NCT02500407) is evaluating mosunetuzumab monotherapy. Data from the Group B cohort, including patients with aggressive or indolent NHL, has been presented [15]. Mosunetuzumab was administered with step-up dosing on days 1, 8 and 15 of cycle 1 followed by a fixed dose on day 1 of each cycle thereafter up to a maximum of 17 cycles. Dosing ranged from a weekly step-up dose of 0.4/1/2.8 mg to 1/2/40.5 mg in the efficacy analysis.

Out of 270 patients, 66.7% had aNHL, including DLBCL, tfFL, MCL and 31.5% had iNHL. There were 30 patients (11.1%) who had received prior CAR T-cell therapy and these included patients with DLBCL, tfFL and FL. In patients with aNHL, the ORR was 37.4% and the CR rate was 19.5% [15]. In patients with iNHL, the ORR was 62.7% and the CR rate was 43.3% [15]. In the patients who had received prior CAR T-cell therapy, the ORR was 38.9% with a CR rate of 22.2% [15]. Of the patients who achieved a CR, the majority remained in remission beyond 12 months off initial therapy.

The latest data from Group B of GO29781 (NCT02500407) detailed the results of mosunetuzumab monotherapy in 62 patients with FL who had received at least two prior systemic therapies [16]. Patients received mosunetuzumab at dose levels from 0.4/1.0/2.8 mg to 1/2/13.5 mg (Cycle 1 Day 1/8/15 dose levels). The ORR was 67.7% with a CR rate of 51.6% [16]. In the 29 patients with progression of FL within 24 months of first treatment (POD24), the ORR was 75.9% with a CR rate of 55.2% [16]. Four patients (6%) had received prior CAR T-cell therapy and all responded with 50% achieving a CR [16]. The median duration of response (DOR) was 20.4 months (95% CI: 9.4–22.7) with a median DOR in patients achieving CR of 21 months (95% CI: 16.0–22.2) [16].

In order to reduce the risk of CRS and investigate alternative dosing strategies, a cohort of 23 patients with R/R NHL in the GO29781 trial (NCT02500407) received subcutaneous (SC) mosunetuzumab monotherapy. Patients with aggressive or indolent NHL were included. Three-weekly doses from 1.6 to 20 mg were assessed. Of the 22 efficacy-evaluable patients, the ORR was 86% and the CR rate was 29% in patients with iNHL and 60% and 20% in patients with aNHL, respectively [17]. After a median of 6.9 months, all but one patient in CR was still in remission [17]. The pharmacokinetic (PK) profile of mosunetuzumab SC showed slow absorption rate and high bioavailability (>75%) with no grade ≥ 2 CRS at doses below 13.5 mg [17].

In the frontline setting, mosunetuzumab is being assessed either as a monotherapy or in combination with chemotherapy. In the phase I/II trial GO40554 (NCT03677154), mosunetuzumab monotherapy is being assessed in patients 80 years or over or in patients with untreated DLBCL between 60 and 79 years of age unsuitable for R-CHOP chemotherapy. Of the evaluable 19 patients, eight patients received the weekly step-up dose of 1/2/13.5 mg whilst 11 patients received the 1/2/30 mg dose level. Treatment was continued up to a maximum of 17 cycles. The median age was 84 (range: 67–100) years. In the 19 patients, the ORR was 58% and CR rate was 42% [18].

Mosunetuzumab is also being evaluated in the upfront DLBCL population who are suitable for CHOP chemotherapy. In the phase Ib/II GO40515 (NCT03677141) study, patients with untreated DLBCL with an IPI of 2 to 5, as well as a cohort with R/R NHL, receive IV mosunetuzumab in combination with CHOP. Of the 43 patients with efficacy data available, seven patients had R/R NHL and 36 patients had newly diagnosed DLBCL. Mosunetuzumab is administered in a weekly step-up dosing regimen in a 1/2/13.5 mg or 1/2/30 mg dose level in R/R NHL or 1/2/30 mg dose level in newly diagnosed DLBCL. In the seven patients with R/R NHL, the ORR was 86% with a CR rate of 71% [19]. In the 27 evaluable DLBCL patients, the ORR was 96% and the CR rate was 85% [19].

### 3.4. Odronextamab

Odronextamab, (REGN1979), is a hinge-stabilised, fully human IgG4-based CD20/CD3 bsAb that has been evaluated in R/R NHL. In a Phase I study (NCT02290951), odronextamab is administered in a step-up dosing regimen over three weeks followed by a fixed weekly dose until week 12. After this, fortnightly maintenance dosing is given. Preliminary data is available for 127 patients with R/R NHL with doses ranging from 0.03 to 320 mg [20]. This cohort included a heavily pre-treated group of patients with DLBCL, grade 1–3a FL and MCL, with 29 patients receiving prior CAR T-cell therapy. 

In patients with R/R grade 1–3a FL, odronextamab, at a dose of ≥ 5 mg (*n* = 28), achieved an ORR of 92.9% and a CR rate of 75.0% [20]. Median DOR was 7.7 months [20]. In patients with DLBCL, excluding those who had received prior CAR T-cell therapy, those treated at doses ≥ 80 mg (*n* = 10), the ORR and CR rate were both 60% [20]. The median observed DOR in the DLBCL group was 10.3 months [20]. In those DLBCL patients who relapsed after CAR T-cell therapy, 21 patients were treated at doses ≥ 80 mg, with an ORR of 33.3% and a CR rate of 23.8% [20].

### 3.5. Epcoritamab

Epcoritamab (GEN3013) is a subcutaneously administered CD20/CD3 bsAb [21]. In the phase I/II trial (NCT03625037), patients with R/R NHL received a SC 1 mL injection of epcoritamab at a flat dose. The dose was administered weekly for two 28-day cycles, fortnightly for four cycles and then four-weekly thereafter. Preliminary data presented outcomes for 67 patients, 45 (67%) with DLBCL, 12 (18%) with FL and 4 (6%) with MCL. Six patients (9%) had received prior CAR T-cell therapy.

In the 18 patients with DLBCL who received epcoritamab at a dose of ≥ 12 mg, the ORR was 66.7% with a CR rate of 33.3% [21]. Of the seven DLBCL patients who received a dose ≥ 48 mg, the ORR was 100% and two patients (28.6%) achieved a CR [21]. In the patients with DLBCL previously treated with CAR T-cell therapy, all responded with two patients achieving a CR and the other two a partial response [21]. In the eight patients with FL receiving a dose ≥ 0.76 mg, the ORR was 100% and two patients achieved a CR [21]. In the four patients with MCL, responses were seen in two patients with blastoid variant MCL [21]. Data on duration of response are not yet available.

### 3.6. IGM-2323

Many of the bsAbs are IgG molecules and alternate designs include an engineered pentameric IgM antibody, IGM-2323. Preliminary data has been presented from the phase I study, NCT04082936, where IGM-2323 has been used in patients with R/R NHL [7]. Eight patients, three with FL, two with MCL, two with marginal zone lymphoma and one with DLBCL, have data available and have been treated at four dose levels (0.5, 2.5, 10 and 30 mg). Two patients have discontinued treatment whilst six have continued on therapy. No drug limiting toxicity or severe adverse events were reported in the patients treated to date.

## 4. Toxicity and Management of Key Adverse Events

Potent T-cell activation induced by bsAbs pose the risk of unique complications such as CRS, immune effector cell-associated neurotoxicity syndrome (ICANS) and tumour flare. Identifying patients at high risk of adverse events and implementing management strategies is important to reduce toxicities and maximising the utility of bsAb. 

Adverse events are common in patients receiving bsAbs, with the majority of patients experiencing grade 3 or higher adverse events (Table 1) [11,19]. Overlapping toxicities are seen with conventional chemotherapy, including fatigue, cytopenias, infections, diarrhea and elevated liver enzymes [10,13,20,22,23]. Tumour lysis syndrome has been reported due to their potent activity, including fatal outcomes [24]. Prophylaxis should be employed for patients at increased risk with high tumour burden, circulating disease, elevated LDH and impaired renal function. Rates of adverse events leading to treatment discontinuation vary, from approximately 5% in clinical trials of mosunetuzumab, to over 20%, predominantly due to severe neurotoxicity, in patients receiving blinatumomab [10,11,15,19]. Grade 5 adverse events are rare [21,22].

### 4.1. Cytokine Release Syndrome

CRS is a common systemic inflammatory response seen in response to antibody and adoptive T-cell therapies, resulting from immune activation and release of inflammatory cytokines [25]. IL-6 is the predominant cytokine most consistently found to be elevated in CRS patients, with high levels correlating with severe CRS [26,27]. Other inflammatory cytokines released from lymphocytes and macrophages, including IL-10, IFNγ and TNFα, are also implicated in the clinical syndrome [28,29]. 

CRS is the most common adverse event seen in many bsAbs used for the treatment of R/R NHL (Table 1). In a phase I trial of glofitamab, 55% of 118 patients with R/R NHL experienced CRS, though only two patients (1.7%) had grade 3 or 4 events [13]. Similar findings of frequent but non-severe CRS have been reported with the other CD20/CD3 bsAb mosunetuzumab and epcoritamab [15,21]. 

Fever within 24 h is the hallmark symptom, with significant variability in other aspects of the clinical presentation. Mild features include fatigue, headache, rash, arthralgia and myalgia. Severe CRS is a medical emergency and can present with circulatory shock, respiratory dysfunction, neurotoxicity and disseminated intravascular coagulation [25]. Intercurrent illnesses and toxicity can complicate the diagnosis. Patients receiving bsAbs are also at risk of sepsis and infection must be investigated and empirically treated in most patients presenting with CRS symptoms. Haemophagocytic lymphohistiocytosis or macrophage activation syndrome, tumour lysis syndrome and hypersensitivity reactions also present with overlapping clinical and laboratory features [30,31].

**Table 1 jpm-11-00355-t001:** Summary of adverse events associated with bsAbs in the treatment of NHL.

		Total AE	Grade ≥ 3 AE	Grade 5 AE (Excluding PD)	AE Leading to Treatment Withdrawal	Pyrexia	CRS	Grade ≥ 3 CRS	Neurotoxicity	Grade ≥ 3 Neurotoxicity	Neutropenia	Grade ≥ 3 Neutropenia
Schuster [15] GO29781	Mosunetuzumab *n* = 270	255 (94%)	170 (63%)	5 (1.9%)	7 (2.6%)	NR	78 (29%)	3 (1.1%)	118 ^a^ (44%)	10 (3.7%)	NR	42 (16%)
Phillips [19] GO40515	Mosunetuzumab + CHOP *n* = 43	NR	37 (86%)	2 (4.6%)	2 (4.6%)	NR	21 (49%)	0 (0%)	NR	NR	25 (58%)	25 (58%)
Dickinson [13] NP30179	Glofitamab + Gpt *n* = 118	NR	NR	NR	NR	41 (35%)	65 (55%)	2 (1.7%)	NR	NR	41 (35%)	NR
Hutchings [22] NP39488	Glofitamab + Gpt + atezolizumab *n* = 43	NR	NR	1 (2.3%)	NR	16 (37%)	16 (42%)	0 (0%)	NR	3 (7%)	9 (21%)	8 (18%)
Hutchings [32] GEN3013	Epcoritamab *n* = 41	NR	NR	0 (0%)	NR	29 (71%)	24 (59%)	0 (0%)	NR	NR	NR	NR
Bannerji [20]	Odronextamab *n* = 127	NR	NR	NR	7 (5.5%)	97 (76%)	79 (62%)	9 (7.1%)	NR	5 (3.9%)	NR	NR
Goebeler [10]	Blinatumomab *n* = 76	NR	NR	2 (2.6%)	17 (22%)	58 (76%)	NR	NR	90 (71%)	17 (22%)	20 (26%)	13 (17%)
Viardot [11]	Blinatumomab *n* = 23	23 (100%)	22 (96%)	1 (4.3%)	5 (22%)	10 (43%)	0 (0%)	0 (0%)	16 (70%)	5(22%)	4 ^b^ (17%)	4 ^b^ (17%)
Coyle [12]	Blinatumomab *n* = 41	41 (100%)	29 (71%)	3 (7.3%)	5 (12%)	10 (24%)	NR	1 (2%)	23 (56%)	10 (24%)	5 (12%)	4 (10%)

AE—adverse event. PD—progressive disease. NR—not reported. Gpt—Obinutuzumab pre-treatment. ^a^ ICANS in three patients (1.1%). ^b^ Indicates rates of leukopenia. Rates of neutropenia not reported.

A number of disparate grading systems for CRS have been developed [25,33,34,35,36]. A consensus grading system aimed at harmonising reporting was developed by the American Society for Transplantation and Cellular Therapy (ASTCT) (Table 2) [30]. The consensus grading system allows better comparison of safety data across different immune effector cell engaging therapies and facilitates the development of optimal and consistent grade-directed management strategies.

Patient, disease and therapy-related factors influence the risk of developing severe CRS. Patients with high disease burden are more likely to experience severe CRS [37,38,39,40]. This finding contributes to the “first-dose effect”, where CRS typically occurs with the first dose of bsAb and rarely occurs with subsequent exposure [13,14,41]. Other risk factors for CRS include the administered dose of therapy, circulating disease and the presence of a pre-existent state of inflammation with either active infection, high ferritin or high C-reactive protein (CRP) [37,42,43,44,45].

Mitigation strategies to reduce severe CRS are commonly used with bsAbs. Premedications are regularly employed with steroids, antihistamines and acetaminophen or paracetamol [8]. Step-up dosing reduces the risk of severe CRS and this approach has been incorporated into ongoing clinical trials of bsAbs in NHL patients [11,46,47]. A novel approach is the use of obinutuzumab 7 days before the first dose of glofitamab [48]. Obinutuzumab partially occupies CD20 antigen epitope targets of glofitamab and profoundly depletes B-cells in the peripheral blood and secondary lymphoid organs, thereby reducing T-cell activation and cytokine release. Use of a bispecific antibody engineered on a pentameric IgM molecular has showed limited and transient cytokine secretion in preclinical data, with IFNγ being the predominant cytokine release rather than IL-6 [49]. Subcutaneous administration of mosunetuzumab has shown reduced IL-6 peak levels and preliminary data show an association with only mild CRS events [17].

Management of CRS follows a grade-adapted strategy. Mild grade 1 CRS is managed with supportive therapy including infusion discontinuation, intravenous fluids, antipyretics and antihistamines. Prolonged or severe CRS requires close monitoring of organ function, generally within an intensive care unit. Tocilizumab, a humanized, IgG, anti-human IL-6R monoclonal antibody is the mainstay of treatment for grade 2 or higher CRS [25]. It is effective at a dose of 8 mg/kg in adults and 12 mg/kg in patients weighing ≤ 30 kg, with a maximum recommended dose of 800 mg per infusion [25,28,30]. A clinical response is generally seen within a few hours and repeated doses can be given if significant improvement does not occur within 8 to 12 h. Of note, monitoring CRP is unhelpful after tocilizumab administration as blockade of IL-6 results in a rapid decrease in CRP [31].

Tocilizumab does not efficiently penetrate the blood–brain barrier and so corticosteroids are used in cases with significant neurotoxicity or co-existing ICANS [25]. Otherwise, corticosteroids are used as second line treatment for severe or refractory CRS. Corticosteroids have widespread downregulatory effect on immune activation, raising theoretical concerns of dampening the immune anti-tumour effect [39,50]. However, tocilizumab and corticosteroid administration to treat CRS do not appear to negatively affect response rates to T-cell engaging therapies [51,52]. Inhibition of other cytokines implicated in CRS can be considered in refractory cases. Use of etanercept (anti-TNFα inhibitor) and anakinra (anti-IL-1R inhibitor) have been reported, though these agents have not been widely used due to the effectiveness of tocilizumab and corticosteroids [25,42,53].

### 4.2. Neurotoxicity and Immune Effector Cell-Associated Neurotoxicity Syndrome

ICANS is a pathological process involving the central nervous system (CNS) following immunotherapy that activates endogenous or infused T-cells [30]. ICANS is commonly associated with CRS and its pathophysiology similarly arises from an enhanced release of proinflammatory cytokines and neurotoxic substances. Endothelial activation and disruption of the blood-brain barrier also plays a critical role in the syndrome [54,55].

Neurotoxicity is common with blinatumomab, occurring in 70% of R/R NHL patients in a phase 2 trial, including 22% with grade 3 or higher toxicity [11]. The toxicity appeared to be dose-dependent and lead to frequent treatment discontinuation (Table 1). Severe neurotoxicity and ICANS are less common with the CD20/CD3 bsAb mosunetuzumab [15,19].

ICANS generally develops within the first seven days of therapy [3]. Headache and tremor are the most common symptoms. Whilst the clinical presentation can vary, it often follows a stereotypic evolution [30]. Tremor, dysgraphia, expressive and nominal dysphasia, impaired attention, apraxia and lethargy are early manifestations. Global aphasia, akinesia, seizures and obtundation are late and severe manifestations. Differentiating ICANS from other neurological conditions can be difficult. Headache is a non-specific symptom, whereas expressive dysphasia is a specific symptom of ICANS and a good predictor of severe neurotoxicity [55].

Similar to CRS, a number of different ICANS grading systems have been developed [33,36]. The ASTCT consensus group formed an Immune Effector Cell-Associated Encephalopathy (ICE) screening tool to score the degree of encephalopathy out of 10 (Figure 1). The ICE score is used together with other features of ICANS in the consensus grading system (Table 3) [30].

Many of the risk factors for CRS also apply to ICANS. Early or severe CRS increases the risk of developing ICANS. High tumour burden, thrombocytopenia, high ferritin and pre-existing neurological co-morbidities are also risk factors [54,56,57]. Most clinical trials exclude patients with CNS disease, so whether CNS involvement increases the risk of developing ICANS remains unclear [58]. Strategies such as pre-medication and step-up dosing used to reduce the risk of CRS also appear effective in reducing severe ICANS [11,46].

Management of ICANS involves thorough and serial investigations with EEG, fundoscopic examination and neuroimaging to exclude other causes of neurotoxicity and to identify further complications. Supportive measures are essential, including elevation of the bed to at least 30°, aspiration precautions, avoiding other CNS depressing medications and mechanical ventilation for airway protection if required. Pharmacological treatment is with high dose corticosteroids (e.g., dexamethasone 10 mg every 6 h or methylprednisolone 1 g/day) followed by a tapered course. Tocilizumab is not used for ICANS management due to its limited ability to cross the blood-brain-barrier, although it is often administered due to the common co-occurrence with CRS [36].

### 4.3. Tumour Flare

Tumour flare is a well described immune reaction in NHL, particularly with exposure to immunomodulators such as thalidomide and lenalidomide [59,60]. A similar phenomenon is seen with T-cell engaging therapies whereby an influx of T-cells into the tumour sites shortly after first administration leads to fevers, tumour pain, lymphocytosis and rash. Other clinical features will vary depending on anatomical location and the size of the tumour. Mass effect can occur on vital structures such as airways, blood vessels and other vital organs. Radiological imaging may show pseudoprogression [61,62].

Dose-limiting tumour flare has been reported with glofitamab when used in combination with atezolizumab, an anti-PD-L1 inhibitor [22]. Prophylactic preventative intervention should be considered in patients with tumors involving critical locations such as the oropharynx and mediastinum. Pain associated with tumour flare can be managed with non-opioid and opioid analgesia, with high dose corticosteroids reserved for persistent or severe cases. 

## 5. Conclusions

Since the approval of blinatumomab for R/R ALL, there has been significant investment into the development of bsAbs across the spectrum of indolent to aggressive lymphomas. The off-the-shelf preparation to provide cell-specific immune engagement has provided a new therapeutic option and, based on early clinical development described in this review, combinatorial and monotherapy bsAb therapy is being tested from frontline through to heavily pre-treated patients, including those who progressed after CAR T-cell therapy.

This review has highlighted some of the mechanistic differences between bsAbs in advanced stage clinical development for lymphoma. Preclinical and biomarker studies including pharmacokinetics and cytokine assays will assist understanding as to the different treatment responses, tumour kinetics and AE profiles of bsAbs. The AE profile is distinct from non-cellular therapies, but relatively uniform to the class and once familiarity with CRS identification, classification and management is achieved, clinicians are well poised to implement the use of bsAbs into widespread clinical practice as safety and efficacy endpoints are met and the likely pathway to approval achieved.

## Figures and Tables

**Figure 1 jpm-11-00355-f001:**
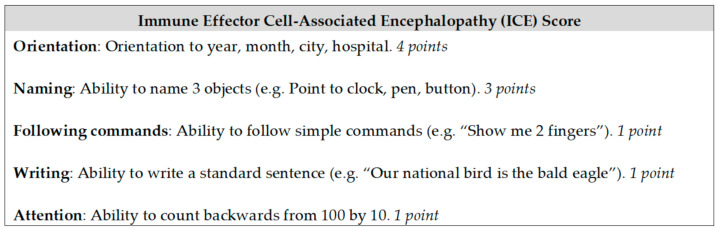
Encephalopathy screening tool–Immune Effector Cell-Associated Encephalopathy (ICE) score, adapted from Lee et al. [30]. Score of 10—no impairment; score of 7–9—grade 1 ICANS; score of 3–6—grade 2 ICANS; score of 0–2—grade 3 ICANS; score of 0 due to patient being unarousable and unable to perform assessment–grade 4 ICANS.

**Table 2 jpm-11-00355-t002:** ASTCT CRS consensus grading, adapted from Lee et al. [30].

CRS Parameter	Grade 1	Grade 2	Grade 3	Grade 4
Fever	Temperature ≥ 38 °C	Temperature ≥ 38 °C	Temperature ≥ 38 °C	Temperature ≥ 38 °C
			With	
Hypotension	None	Not requiring vasopressors	Requiring a vasopressor, with or without vasopressin	Requiring multiple vasopressors, excluding vasopressin
			And/or ^a^	
Hypoxia	None	Requiring low-flow ^b^ nasal cannula or byblow	Requiring high-flow ^b^ nasal cannula, facemask, nonrebreather mask, or Venturi mask	Requiring positive pressure

^a^ CRS grading is determined by the most severe event. ^b^ Low-flow defined by oxygen delivered at ≤6 L/min. High flow defined by oxygen delivered at >6 L/min.

**Table 3 jpm-11-00355-t003:** ASTCT ICANS Consensus Grading, adapted from Lee et al. [30].

Neurotoxicity Domain	Grade 1	Grade 2	Grade 3	Grade 4
ICE score	7–9	3–6	0–2	0 (patient is unarousable and unable to perform ICE)
Level of consciousness	Awakens spontaneously	Awakens to voice	Awakens only to tactile stimulus	Patient is unarousable or requires vigorous/repetitive stimuli to arouse. Stupor or coma
Seizure	N/A	N/A	Any clinical seizure or nonconvulsive seizures on electroencephalogram (EEG) that resolve with intervention	Life-threatening prolonged seizure > 5 min, or repetitive clinical/electrical seizures without return to baseline in between
Motor findings	N/A	N/A	N/A	Deep focal motor weakness such as hemiparesis or paraparesis
Elevated ICP or cerebral oedema	N/A	N/A	Focal/local oedema on neuroimaging	Diffuse cerebral oedema on neuroimaging, decerebrate/decorticate posturing, cranial nerve VI palsy, papilloedema, Cushing’s triad

ICANS grade is determined by the most severe event not attributable to any other cause. ICE score of 0 may be classified as grade 3 ICANS if awake with global aphasia, or grade 4 if unarousable. Tremor, myoclonus and intracranial hemorrhage are not features of ICANS grading.

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
