# Peer review of "Bispecific Antibodies: A Review of Development, Clinical Efficacy and Toxicity in B-Cell Lymphomas"

_jpm, 2021, doi:10.3390/jpm11050355_

Round 1

Reviewer 1 Report

Article ''Bispecific Antibodies: A Review of Development, Clinical Effi-2 cacy and Toxicity in B-Cell Lymphomas''  has an interesting and practical, but it also needs changes. In general, it is better to express sentences in a more understandable way.The method of the study is not well described in the abstract. Writing problems such as not observing a suitable place for distance and point, etc. were observed. It is better to state the discussion part first and then finally the conclusion is expressed.

Reviewer 2 Report

This is a timely review on an emerging and rapidly expanding topic, the use of bispecific antibodies in NHL. Data are still preliminary, often only reported in abstract form, but the review will help the reader to approach this field. The manuscript is well written and presents data in a balanced manner.

Some minor poins:

It is not clear from chapter 2.3 whether Fc modifications in bs Ab are intended to reduce binding to Fc gamma recptors and complement (lines 96,97), or to enhance ADCC and CDC activation (lines 108,  117)

Line 108: Obinutuzumab was engineered to enhance ADCC, not CDC.

Line 117: please confirm that you mean CDC and not ADCC
